# Construction and Mechanism Analysis of a Self-Assembled Conductive Network in DGEBA/PEI/HRGO Nanocomposites by Controlling Filler Selective Localization

**DOI:** 10.3390/nano11010228

**Published:** 2021-01-16

**Authors:** Yiming Meng, Sushant Sharma, Wenjun Gan, Seung Hyun Hur, Won Mook Choi, Jin Suk Chung

**Affiliations:** 1School of Chemical Engineering, University of Ulsan, Daehakro 93, Namgu, Ulsan 44610, Korea; mengyiming1112@gmail.com (Y.M.); sushant.nplindia@gmail.com (S.S.); shhur@mail.ulsan.ac.kr (S.H.H.); wmchoi98@ulsan.ac.kr (W.M.C.); 2Department of Macromolecular Materials and Engineering, College of Chemistry and Chemical Engineering, Shanghai University of Engineering Science, Shanghai 201620, China; wjgan@sues.edu.cn

**Keywords:** polyblend, nanocomposites, reduced graphene oxide, curing reaction-induced phase separation (CRIPS), selective localization, microstructural analysis

## Abstract

Herein, a feasible and effective approach is developed to build an electrically conductive and double percolation network-like structure via the incorporation of highly reduced graphene oxide (HRGO) into a polymer blend of diglycidyl ether of bisphenol A/polyetherimide (DGEBA/PEI). With the assistance of the curing reaction-induced phase separation (CRIPS) technique, an interconnected network of HRGO is formed in the phase-separated structure of the DGEBA/PEI polymer blend due to selective localization behavior. In this study, HRGO was prepared from a unique chemical reduction technique. The DGEBA/PEI/HRGO nanocomposite was analyzed in terms of phase structure by content of PEI and low weight fractions of HRGO (0.5 wt.%). The HRGO delivered a high electrical conductivity in DGEBA/PEI polyblends, wherein the value increased from 5.03 × 10^−16^ S/m to 5.88 S/m at a low content of HRGO (0.5 wt.%). Furthermore, the HRGO accelerated the curing reaction process of CRIPS due to its amino group. Finally, dynamic mechanical analyses (DMA) were performed to understand the CRIPS phenomenon and selective localization of HRGO reinforcement. The storage modulus increased monotonically from 1536 MPa to 1660 MPa for the 25 phr (parts per hundred in the DGEBA) PEI polyblend and reached 1915 MPa with 0.5 wt.% HRGO reinforcement. These simultaneous improvements in electrical conductivity and dynamic mechanical properties clearly demonstrate the potential of this conductive polyblend for various engineering applications.

## 1. Introduction

In the 21st century, considering the global environment, energy efficiency and renewable energy technologies, the core elements for many such fields are polymer reinforced composites and their derived products. With the rapid growth in development and usage of electronic and electrical equipment, electronic packaging of polymers has become critical. Therefore, need for affordable multi-functional polymer materials has increased significantly.

Epoxies are highly cross-linked thermosetting polymers. One common epoxy is diglycidyl ether of bisphenol A (DGEBA), which possesses excellent corrosion resistance, outstanding adhesion strength, low curing shrinkage, and excellent mechanical properties. DGEBA is durable in harsh environments and dimensionally stable. Hence, it is widely used in coatings [1,2], adhesives [3], supporting structural materials [4], automotive electron devices [5], and Electromagnetic interference shielding [6], several other engineering applications [7,8]. However, this kind of structure has a few drawbacks that lead to brittle failure, low impact resistance, and poor conductivity, which limit extensive use of DGEBA epoxy resin in various engineering applications. To enhance the properties of the pristine epoxy resin, researchers generally blend it with thermoplastic polymers [9], rubber [10], or polysiloxane [11] as a modifier that can enhance energy dissipation mechanisms in the matrix. These modifications result in appreciable improvement in the toughness of the epoxy resin. However, this often causes a significant decrease in epoxy resin tensile strength, modulus, and maximum performance temperature by lowering the glass transition temperature (T_g_) and does not contribute to electrical conductivity [12]. Higher energy absorption at a maximum working temperature along with electrical conductivity are important for a broad range of engineering applications.

Therefore, many nanofillers such as carbon nanotubes [13], graphene and derivatives [14], carbides [15], nitrides [16], and hybrids [17,18,19] have been extensively used to introduce electrical properties in polymers due to their inherited electrical conductivity and high aspect ratios [20,21]. Several parameters affect the electrical conductivity of polymer nanocomposites, such as interfacial resistance between filler and matrix, size and type of filler, processing conditions, and localization of filler in the matrix system. It is quite challenging to maintain a well distributed network of nanofillers in a polymer matrix, which is necessary for improving electrical conductivity. Generally, these nanofillers experience strong van der Waals forces at loadings of 1–5 wt.% reinforcement, which reduce their dispersion. Various strategies have been adopted in recent years to establish a uniform network of nanofillers in polymer composites, such as segregating the structures of polymer nanocomposites by compression molding [22,23,24] and reinforcing macroscopic scaffolds of nanofillers network in a polymer matrix [25,26,27]. However, these polymer composites are not categorized as economical multi-functional as their processing involves multiple time-consuming steps.

Another technique is selective localization of nanofillers in an immiscible polyblend with a dispersed network using a curing reaction-induced phase separation technique [28,29,30]. Double percolation was initially observed by Sumita et al. [31] in carbon black (CB)-filled immiscible polymer blends. Thereafter, the double percolation concept was extensively applied in carbon-based materials such as graphene nanoplatelets (GNPs) [32], functional graphene oxide, carbon nanotubes (CNTs) [33], and carbon fiber (CFs) [34]. Physical mixing of polymer blends (polyblends) can be divided into the following methods [35]: (i) mechanical mixing in rubber mills or extruders, (ii) polymerization of one monomer in the presence of another, (iii) evaporation or precipitation from a mixture of polymer solutions, and (iv) coagulation of a mixture of polymer lattices. However, polymerization in the presence of another polymer is an economical and easily controllable technique to form a low-percolation conductive network of carbon nanofillers in a multiphase polyblend. For example, Zhang et al. [36] formed a co-continuous network of epoxy (EP)/ polyether sulfone (PES)/ multiwalled carbon nanotubes (MWCNTs) and selectively localized the MWCNTs in an epoxy-rich area using reaction-induced phase separation and filler localization. In the experimental study, it was found that after 20 wt.% of PES mechanical properties of polyblend increases along with high glass transition temperature (T_g_). With the selective localization of MWCNTs network (4.8 wt.%) in polyblend, the electrical conductivity increased by 450% compared to that without a PES polyblend. Luna et al. [37] studied the effects of nanoparticles on the morphologies of various immiscible polyblends and analyzed their effect of electrical properties of polyblend nanocomposites. The analytical study confirmed that reinforcement of nanofillers ensure a low thermal expansion coefficient, high thermo-mechanical resistance, and superior electrical conductivity if the blend exhibited a co-continuous microstructure in the fillers [38]. Huang et al. [28] used the double percolation technique to control selective localization of CNTs. They prepared a co-continuous immiscible polyblend of poly (lactic acid)/poly(ε-caprolactone) (PLA/PCL) at a weight ratio of 50/50 and controlled the migration of MWCNTs from unfavorable PLA to favorable PCL phase. As a result, percolation of PLA/PCL/MWCNT composites (0.97 wt.%) decreased by two orders of magnitude in PLA/MWCNT/PCL composites (0.025 wt.%) for similar electrical conductivities. Hence, development of a conductive composite is recommended with low conductive filler weight ratios and a combination of dissimilar polyblends [39,40,41]. When the polyblend system contains a thermosetting component, it is very difficult to control selective localization of conducting nanofillers at the interphase. Generally, the fillers become concentrated in a particular phase depending on the effect of the curing reaction on miscibility, which ultimately increases the percolation threshold via dominance of a particular phase.

Hence, a facile approach was developed in this study to construct a conductive double percolation network of highly reduced graphene oxide (HRGO) in a polyblend of diglycidyl ether of bisphenol A/polyetherimide (DGEBA/PEI). In this technique, conductive HRGO filler was introduced into DGEBA/PEI two-phase immiscible polyblend and was selectively localized at the interphase or in the co-continuous phase. The DGEBA/PEI/HRGO nanocomposite showed high electrical conductivity with a low concentration of nanofiller (i.e., 0.5 wt.%) because of its double percolation structure. Furthermore, the structural and morphological analyses of DGEBA/PEI/HRGO nanocomposites were conducted using a field emission scanning electron microscope (FESEM) and an inverted optical microscope (OM) to characterize the conductive network. The electrical conductivity of the nanocomposites was measured by using a four-point probe technique. The effects of HRGO on the CRIPS behavior of the composites were analyzed by dynamic mechanical analysis (DMA), and the influence of HRGO on the storage modulus and T_g_ of the DGEBA/PEI/HRGO was analyzed in detail. This ergonomic approach and the ability to control the localization of a nanofiller network are helpful in developing an economical multifunctional polyblend of DGEBA/PEI/HRGO that can be used in various engineering applications such as electronic coatings, packaging, and electromagnetic shielding.

## 2. Experimental Section

### 2.1. Materials

Commercial graphene oxide was supplied by Standard Graphene, Ulsan, Korea. Diglycidyl ether of bisphenol A (DGEBA, DER 332) was provided by Dow Chemical Co. ( Midland, MI, United States of America) with an epoxide equivalent weight of 171–175 g/eq. The curing agent methyl tetrahydrophthalic anhydride (Me-THPA) and accelerator N, N-dimethylbenzylamine (DMBA) were obtained from Shanghai Reagent Co., China. Polyetherimide (PEI, Ultem^@^1000, T_g_ ≈ 217 °C) was purchased from Goodfellow Company (Huntingdon, United Kingdom). N, N-Dimethylformamide (DMF), ethanol (EtOH), and methylene dichloride (CH_2_Cl_2_) were purchased from Sigma-Aldrich (Louis, MO, United States of America). All reagents were used without further purification.

### 2.2. Synthesis of Highly Reduced Graphene Oxide (HRGO)

The HRGO was prepared according to the method of Dang et al. [42], with some adjustments. For this, 2000 mg graphene oxide (GO, Standard Graphene, Korea) was added into a mixed solution of 800 mL N, N-dimethylformamide (DMF) and 200 mL de-ionized water (DIW). The mixture was sonicated in an ultrasonic bath (Jeiotech UC-10, 200 W, Seoul, Korea) for 3 h to form a homogenous suspension of GO (2 mg/mL). Hydrazine reduction was achieved by adding 5 mL of hydrazine monohydrate (98%) per 100 mL of GO suspension and stirring at 60 °C in an oil bath for 24 h. The HRGO suspensions were filtered and washed with large amounts of ethanol to remove excess hydrazine monohydrate. The HRGO filter cakes were re-dispersed in ethanol by sonication for 3 h (the temperature of the sonication bath was maintained below 30 °C).

To determine the dispersibility of HRGO in ethanol, 50 mL of HRGO suspension was centrifuged at 3000 RPM for 15 min. Then, 20 mL of the upper supernatant was collected, coagulated by adding a few drops of HCl (1 M), and filtered. The filter cakes were washed with ethanol for three times, dried in vacuum at 100 °C, and weighed to calculate the dispersibility of the HRGO.

### 2.3. Preparation of DGEBA/PEI/HRGO Composites

Composites with and without HRGO were prepared by using following process. First, DGEBA, PEI, and HRGO were weighed according to the formulations in Appendix A and DGEBA/PEI/HRGO composites were denoted by DPxH. Where D, P, x, and H were representing the DGEBA, the PEI, PEI parts per hundred in the DGEBA (phr), and 0.5 wt.% of HRGO, respectively. PEI was dissolved in methylene dichloride (CH_2_Cl_2_) with strong magnetic stirring and was introduced into the HRGO solution via magnetic stirring at room temperature. After that, the DGEBA was added to the PEI/HRGO blends with high-speed magnetic stirring at 80 °C for 2 h to remove the solvent from the mixture. Then, the blends of DGEBA/PEI/HRGO were placed in a vacuum oven at 120 °C for 12 h to degas the mixtures and to remove the solvents. The mixture of curing agent Me-THPA and accelerator DMBA at 400:1 ratio by weight (C-A solution) was added into the vacuum-removed blends and stirred at 550 RPM and 120 °C. The homogeneous blend was achieved by mixing, and the few air bubbles caused by these processes were removed through vacuum treatment at 120 °C for 10 min. Finally, the polyblends were pre-curing at 150 °C for 5 h and continued by post-curing at 200 °C for 2 h after pouring into the pre-prepared mold (Figure 1). The residual samples in the beaker were stored in a refrigerator for OM measurement. For comparison, neat DGEBA, DP5, DP10, DP15, DP20, DP25, and DP30 were prepared by the same process.

### 2.4. Measurement and Characterization

Optical microscopy (OM) measurements were performed on an inverted metallurgical microscope (DMI3000B, Leica, Seoul, Korea) to observe the phase separation behavior and double percolation structure. Scanning electron microscope (SEM) (S-3400N, Hitachi High-Technologies, Chicago, IL, United States of America) analyses were conducted to observe the morphology of the HRGO. Transmission electron microscopy (TEM) measurements were performed on a field emission transmission electron microscope (Tecnai G2 F20 X-Twin, FEI Co., Hillsboro, OR, United State of America) operated at an acceleration voltage of 200 kV. The ultrathin film (thickness:100–200 nm) composite samples were prepared at room temperature using an ultramicrotome (RMC CR-X, Boeckeler Instruments, Tucson, AZ, United States of America) equipped with a glass knife. The volume resistivity and volume conductivity of nanocomposites were measured at room temperature using a four-point probe technique with (CMT-100, AiT Co., Gyeonggi, Korea). Nanocomposites were prepared in a disk shape (approximately 40 mm diameter and 2 mm thickness) for conductivity measurements (Figure 1). The thermomechanical properties of the nanocomposites were performed by dynamic mechanical analysis (DMA) (TA Q800, TA instrument, New Castle, DE, United States of America) in single cantilever mode at a frequency of 1 Hz and oscillation strain of 0.2. The temperature ranges were 30–180 °C and 30–230 °C at a heating rate of 3 °C/min for the neat DGEBA and other nanocomposites, respectively. The samples were cast in square aluminum molds, yielding a specimen geometry of 45 mm × 10 mm × 3 mm (Figure 1). The contact angle (CA) measurements of DGEBA, PEI, and HRGO films with various liquids were measured using a (Phoenix 300, SEO, Komachine Co., Seoul, Korea). The CA measurements were conducted using the sessile drop method at room temperature with 15–17 μL volume drops of liquids that were prepared with a micro syringe. The surface tension and polarity of deionized water, glycerol, and formamide are shown in Appendix A. The Raman spectra were characterized using a confocal Raman microscope (Thermo Scientific, Seoul, Korea) with a 532 nm wavelength monochromatic excitation laser with 2 mW laser power and a 5 s exposure time. Furthermore, Fourier transform infrared (FT-IR) spectra were recorded using (FT-IR, Thermo Electron Co., Waltham, MA, United States of America). Differential scanning calorimeter (DSC) measurements were performed by (TA Q20 V24.10, TA instrument, New Castle, DE, United States of America).

## 3. Results and Discussion

### 3.1. Physical and Morphological Properties of HRGO

A low-magnification SEM micrograph of an HRGO flake is shown in Figure 2a. GO is a 2D sheet-like structure and contains multiple lamellar layers that open into a few layers with improved graphitic structure after reduction. Figure 2a represents the opened flakes of graphene layers with irregular and folding which is caused due to acid treatment. They are entangled with each other and helpful in creating a dispersed network by selective localization while reinforcing the polyblend. Figure 2b shows a TEM image of single- or few-layer HRGO nanosheets with many wrinkles. These kinds of 2D wrinkled structures are useful in improving the reinforcement efficiency of HRGO. However, it is very difficult to predict the actual changes in structure after reduction from these SEM and TEM analyses. Therefore, Raman analysis was also conducted on these before reinforcement.

Raman spectroscopy provides information based on inelastic scattering of a molecule irradiated by a monochromatic light, where laser is normally used. Figure 2c shows Raman spectra for GO and HRGO, with two fundamental vibrations observed in the range of 1100–1700cm^−1^. The D vibration band (D-band) was observed at 1345.9 cm^−1^ and 1344.008 cm^−1^ for GO and HRGO, respectively. On the other hand, the G vibration band (G-band) appeared at 1590.8 cm^−1^ and 1586.5 cm^−1^ for GO and HRGO, respectively. Furthermore, the G vibration band is affected by the presence of the stretching C–C bond, which is common in all sp^2^ carbon systems. The D band and G band in the Raman spectra symbolize disorder bands and tangential bands, respectively. Besides, a broadening and shift to higher wavenumbers of the 2D band were observed at 2684.2 cm^−1^ and 2673.6 cm^−1^ for GO and HRGO, respectively. The 2D band can be used to determine number of layers of graphene (monolayer, double layer, or multilayer) as it is highly sensitive to the stacking of graphene layers. In addition, the shifted location of the 2D band is due to the presence of oxygen-containing functional groups and prevents graphene layer stacking of HRGO. Thus, fewer oxygen-containing functional groups remained, allowing the RGO to stack. The I_D_/I_G_ ratio for GO was 0.847. After reduction, the ratio for HRGO increased to 1.063 due to restoration of sp^2^ carbon and decrease in the average size of sp^2^ domains upon reduction [43,44]. The higher intensity in the D band also suggested that more isolated graphene domains were present in HRGO compared to GO due to removal of oxygen moieties from GO after hydrazine reduction [45,46]. Further, to observe the reduction state of HRGO, FT-IR analysis was also conducted. Appendix A represents the FT-IR spectra of GO and HRGO, which confirms the reduction of GO after hydrazine treatment. After reduction of GO, a new amino group from hydrazine was found in HRGO which helped accelerating the curing reaction process.

### 3.2. Morphology of DGEBA/PEI/HRGO as Observed by Optical Microcopy(OM) Measurement

Curing reaction-induced phase separation (CRIPS) is commonly described as separation of a homogeneous polyblend into two immiscible phases during curing. The morphologies of the fabricated DGEBA/PEI/HRGO nanocomposites with different contents of PEI were characterized by OM measurement (Figure 3). During the CRIPS process, the polyblend followed a spinodal decomposition mechanism [47]. The DGEBA/PEI polyblend is a dynamic asymmetric system that initially is homogeneous. The molar mass of the DGEBA increases as the isothermal curing reaction proceeds, resulting in an increase in cross-linking density of DGEBA. Because large differences in mobility occur between DGEBA molecular and PEI, the PEI is no longer miscible in the matrix, and PEI begins to separate out, causing phase separation. The viscoelastic effect become more prominent after primary phase separation, and this reduces the mobility of polymer chains of each polymer as they diffuse into each other. When the diffusion is too slow to achieve geometrical coarsening, the local concentration equilibrium is disrupted, and secondary phase separation will occur in both DGEBA-rich and PEI-rich domains [48].

Figure 3a–g shows the final phase of the DGEBA/PEI/HRGO nanocomposites with various PEI contents. The final phase structure changed from a Figure 3a-like insulated island dispersed phase structure to a Figure 3e micro-size co-continuous phase structure, with a PEI concentration from 5 to 25 phr of DGEBA, respectively. Furthermore, an inversion phase structure (PEI-rich phase as matrix) appeared in the polyblend with increased concentration of PEI, as depicted in Figure 3f.

The PEI-rich phase is represented as darker domains, and the DGEBA-rich phase as brighter domains (Figure 3f). Further these separated phases were confirmed by FETEM analyses in upcoming section. Occurrence of secondary phase separation, which shows many small droplets in both domains at the final stage of phase separation, as in Figure 3e–g, concur well with the state of viscoelastic phase separation reported by Tanaka et al. [49].

Compared with the non-HRGO-filled polyblend in Figure 3g, introduction of the HRGO (Figure 3f) both suppressed the extent of coarsening (i.e., the sizes of the DGEBA-rich phases became smaller) and accelerated the curing process due to interactions between -NH_2_ functional groups on the HRGO and DGEBA oligomer [42]. HRGO served as a crosslinking accelerator as a chemical crosslinking point.

### 3.3. Field Emission Transmission Electron Microscopy (FETEM) Analyses

The distribution of HRGO in DGEBA/PEI polyblend system and the phase structure of polyblend were analyzed by the FETEM technique as well. Figure 4 represents the FETEM micrographs of the DP25H and DP30H polyblends, in which dark and white domains represents PEI and DGEBA phase, respectively. Figure 4a shows a co-continuous phase morphology in low magnification which is consistent with the optical micrographs and Figure 4b shows the crumpled surfaces (with some wrinkles) of sheets of HRGO at the interphase (between DGEBA and PEI phase). It is implied that the HRGO sheets were selectively localized at the interphase in DP25H. Then, the high-resolution transmission electron microscopy (HRTEM) image of DP25H exhibits fringes from which the d-spacing value of the HRGO was calculated to be 0.376 nm as depicted in Figure 4c. Figure 4d–f represents a typical phase inversion structure of DP30H polyblend from lower to higher magnifications. The secondary phase separation structure showed in Figure 4e, involved many dark insulated islands (PEI phase) and numerous crumpled HRGO sheets distributed at the interphase that between DGEBA and PEI phase.

Comparing Figure 4a with Figure 4d, we observe the formation of different phase structures by increasing the content of PEI from 25 to 30 phr. Meanwhile, the FETEM analysis is cohort with the OM micrographs and represents localization of HRGO at the interphase between DGEBA and PEI successfully.

### 3.4. Electrical Properties of Nanocomposites

Morphologic analyses of DP25H and DP30H demonstrated the existence of a co-continuous phase and an inversion phase, respectively. HRGO was selectively located at the interphase between DGEBA and the PEI phase and formed a double-percolation conductive network structure. This dispersed network in polyblend is useful in introducing conductivity to a composite system. Figure 5a,b illustrates the effects of PEI content and PEI/HRGO content on the in-plane conductivity of polyblend and HRGO-reinforced polyblend composites, respectively. Figure 5b shows that the electrical conductivities gradually increased by increasing the PEI content when HRGO was introduced into the DGEBA/PEI polyblend system. On the other hand, there was little change in the electrical conductivity of polyblend without HRGO, as represented in Figure 5a.

However, the phase morphologies of the nanocomposites, especially phase continuity, play an important role in establishing a conductive network of filler in polymer composites. The volume resistivity and conductivity of DGEBA with similar concentrations of PEI and 0.5 wt.% HRGO are presented in Figure 5b and Table 1. With HRGO reinforcement in DGEBA, the conductivity reached 1.83 S/m, and it increased with respect to PEI concentration increases. In the DP25H, the conductivity reached ~5 S/m, an overall improvement of 173% compared to that without PEI. After reaching 25 phr of PEI, the conductivity remained nearly constant, but an inversion phase structure affected the thermomechanical properties of the overall polyblend system. Furthermore, the comparison of electrical properties with other previously reports are shown in Table 2.

The simultaneous blending of PEI with the curing reaction causes phase separation due to a spinodal shift, allowing formation of a co-continuous network structure. This network improves the dispersion state of HRGO and improves the conductivity. This conducting property of nanocomposites is determined by selective localization of filler particles in the polyblend. There are various thermodynamic strategies to control selective localization of filler. The most widely used thermodynamic parameters for localization are the wetting parameters, which are described in an upcoming section [36].

### 3.5. Prediction for Selective Localization of HRGO

Various studies of different complex parameters have been conducted for localization of a filler in an immiscible polyblend. Accurate analysis of localization is difficult because, in addition to thermodynamics, the fluid dynamics, surface tension, and polarity of constituent blends during reaction are very important [56,57,58,59,60,61]. However, still wettability parameter is a useful technique as it considers the surface tension of each component in the blend. To precisely evaluate the surface tension of DGEBA, PEI, and HRGO, we conducted contact angle measurements in conjunction with the Lifshitz–van der Waals/acid-base approaches (Equations (1) and (2)) [56,57]. The values of these surface tension components and parameters of the test liquids used in this work are shown in Table 3. Digital images of the contact angles and values of three components are shown in the Appendix A.
(1)$$\gamma_{L}\left(1+\cos\theta \right)=2\left(\sqrt{\gamma_{s}^{\mathrm{Lw}}\gamma_{L}^{\mathrm{Lw}}}+\sqrt{\gamma_{s}^{+}\cdot \gamma_{L}^{-}}+\sqrt{\gamma_{s}^{-}\cdot \gamma_{L}^{+}}\right)$$
(2)$$\gamma_{L}^{\mathrm{AB}}=2\sqrt{\gamma_{L}^{-}\cdot \gamma_{L}^{+}}$$

The Lifshitz–van der Waals/acid-base approaches proposed by Van Oss et al. [56,57] combine dispersion (γ^d^), polar (γ^P^), and γ^i^ components into a single component, called the nonpolar or Lifshitz–van der Waals component (γ^LW^). Also, the electron acceptor-electron donor (Lewis acid/base) interactions of polar composites are expressed as γ^AB^, and the surface tension of compound i (L: liquid, S: solid) is expressed as γ_i_^+^ (acidic component) γ_i_^−^ (basic component) according to Equation (2). Eventually, the total surface tension is obtained by addition of the nonpolar and polar components (γ^tot^ = γ^LW^ + γ^AB^). Combining this method with the Young-Dupré equation yields Equation (1).

Since Equation (1) contains three unknowns of the solid (γ_S_^LW^, γ_S_^+^, and γ_S_^−^), we used three different liquids (in Table 3), two of which were polar. In our work, the surface tensions of DGEBA, PEI, and HRGO were calculated by Lifshitz–van der Waals approaches as shown in Table 4.

The surface tension of each component in the polyblend is considered essential for predicting the localization. From a thermodynamics perspective, the wetting coefficient (ω_a_) proposed by Sumita et al. [62] was widely used to forecast the localization of nanofillers [51,63,64]. The wetting coefficient ω_a_ can be calculated according to Young’s equation (Equation (3)) as given below:(3)$$\omega_{a}=\frac{\gamma_{\mathrm{HRGO}-B}-\gamma_{\mathrm{HRGO}-A}}{\gamma_{A-B}}$$
where γ_HRGO-B_ is the interfacial tension between HRGO and the polymer B phase, γ_HRGO-A_ is the interfacial tension between HRGO and the polymer A phase, and γ_A−B_ is the interfacial tension between the polymer A phase and polymer B phase. The prediction is that HRGO will be distributed preferentially in the polymer B phase if ω_a_ < −1, HRGO will be located at the interphase between polymer A and polymer B if −1 < ω_a_ < 1 and HRGO will be distributed in the polymer A phase if ω_a_ > 1.

The interfacial tension between the two phases of nanocomposites, γ_A−B_, can be calculated using Wu’s harmonic mean average [65] as follow:(4)$$\gamma_{A-B}=\gamma_{A}+\gamma_{B}-4\left(\frac{\gamma_{A}^{d}\gamma_{B}^{d}}{\gamma_{A}^{d}+\gamma_{B}^{d}}+\frac{\gamma_{A}^{P}\gamma_{B}^{p}}{\gamma_{A}^{P}+\gamma_{B}^{P}}\right)$$
where γ_A−B_ is the interfacial tension between phases A and B. γ_A_ and γ_B_ are the surface tensions of phases A and B, respectively. γ^d^_A_ and γ^d^_B_ are the surface tensions of dispersion components A and B, respectively. γ^p^_A_ and γ^p^_B_ are the surface tensions of polar components A and B, respectively, for which γ = γ^d^ + γ^p^.

In our DGEBA/PEI/HRGO polyblend system, the surface tensions of components are shown in Table 4, and the interfacial tension and calculated wetting coefficient are given in Table 5. From these calculations −1 < ω_a_ = 0.171 < 1, indicating that the HRGO selectively localizes at the interphase in DGEBA/PEI/HRGO systems. This prediction is consistent with the OM images.

### 3.6. Mechanism of HRGO Localization

Considering the final electrical properties of the ternary nanocomposites, selective localization of nanofiller in a polyblend is the key factor for controlling these properties. Introducing the electrical conductivity with very low filler content is very important. In creation of polyblend, crosslinking of DGEBA begins once the temperature increases to the curing temperature, and this is the starting point for phase separation [66,67]. As curing continues, the spinodal downshift led to formation of a co-continuous phase structure. The HRGO nanoparticles were uniformly mixed in the polyblend (Figure 6a). During curing of DGEBA, an island-like structure is formed and progressively increases until the dynamic forces between DGEBA and PEI are in balance. Here, HRGO nanoparticles due to double percolation formed a conductive network at the interphase of DGBEA and PEI, as represented in Figure 6a,b.

Figure 6b presents the TEM micrograph of DP25H nanocomposites in which the two phases of DGEBA and PEI are represented by yellow and orange color dots, respectively. The HRGO particles are clearly visible at the interphase of the two polymers, forming a conductive network in the polyblend at a very low concentration. Table 2 compares our work with previous literature, showing high electrical properties in comparison with others work. Moreover, the content ratio of nanofiller HRGO in our work is very low (only 0.5 wt.% of filler) for achieving the similar or more electrical conductivity compared to others work.

### 3.7. Dynamic Mechanical Analyses (DMA)

The viscoelastic behavior of HRGO-reinforced polyblends and selective localization of HRGOs in nanocomposites have been studied using dynamic mechanical analysis (DMA). DMA provides a sinusoidal load over the clamped sample and compares this with the collected response using a linear variable differential transformer (LVDT). Specifically, the LVDT measures the amplitude of the resulting sinusoidal wave, which is the storage modulus (E’) of the material, and the tangent of the phase lag between applied force and material response (tan δ) [13].

Figure 7a–d presents the storage modulus versus temperature curve of the DGEBA/PEI polyblend with various phr of PEI and that of the DGEBA/PEI/HRGO nanocomposite with various phr of PEI and a fixed concentration of HRGO (0.5 wt.%). Figure 7a presents the storage modulus of a polyblend without HRGO. The storage modulus of neat DGEBA is 1536 MPa in the glassy region (measured at 30 °C) and 14.2 in the rubbery region (measured at T_g_ + 30 °C). The storage modulus increased with PEI and reached maximum values of 1660 MPa and 68 MPa in the glassy and rubbery regions, respectively, for 25 phr PEI. The storage modulus of DP25 was higher for the entire working temperature range (30–180 °C), which suggests that restriction of polymer chains in the polyblend is primarily due to phase separation but also to arrested motion of the thermoplastic monomer chain between DGEBA-rich zones. Figure 7b shows curves of the storage modulus in the temperature range of 80–130 °C, and it clearly indicates the restriction of polymer chain motion at the beginning of the transition zone (glassy to rubbery). Ultimately, the immiscible polymer blending resulted in a binary polymer system that is capable of withstanding higher temperatures than that of DGEBA alone.

Figure 7c and Table 6 show the effect of reinforcement of HRGO on the thermomechanical properties of DGEBA/PEI/HRGO nanocomposites with different phr fractions of PEI and a fixed weight ratio of HRGO. The storage moduli of ternary polyblend nanocomposites increased with increasing PEI concentration. The storage modulus of DP25H reached 1915 MPa and 64.8 MPa in the glassy and rubbery regions, respectively, and showed an overall improvement of ~14% and ~222% over the baseline DGEBA nanocomposite (i.e., without polyblend DP0H). The HRGO reinforcement in polyblend played a crucial role in improving the thermomechanical properties, as depicted in Figure 7d. Although the storage modulus is high for the entire temperature range in DP25H, the effect is more prominent at the beginning of transition (i.e., 80–130 °C), where there was a significant difference in the storage.

Tan δ versus temperature curves of DGEBA/PEI polyblend and DGEBA/PEI/HRGO nanocomposites are given in Figure 8. The curves in Figure 8a–c demonstrate that the glass transition temperature (T_g_) increased with blending concentration of PEI. The T_g_ of base DGEBA was 123.1 °C. After blending with PEI, it increased to 133.8 °C for DP25, an improvement of 10.7 degrees (T_g1_ in Table 6). On the other hand, the T_g_ of neat PEI was 217 °C, which was reduced after blending with DGEBA (T_g2_ in Table 6). The T_g2_ of the DP25 was 192 °C. When HRGO was added to the polyblend, the T_g2_ showed similar results (Figure 8d–f). The T_g1_ of DP25H reached 131.6 °C, which is close to the T_g1_ of DP25 but higher than that of DGEBA/0.5HRGO (123.3 °C). The T_g2_ of DP25H was 198.2 °C, higher than that of DP25 (192.0 °C). This implies that addition of HRGO in the polyblend restricts polymer chain motion of both DGEBA and PEI. However, it is more prominent for PEI, as it exists only at the narrow zones between the large islands of DGEBA created during phase separation. The decrease in peak height of tan δ in polyblend (Figure 8b) and nanocomposite (Figure 8e) indicates the change in internal energy due to restriction of molecular mobility caused by the increase in viscosity due to PEI and HRGO fillers, respectively [68,69].

Further, to investigate the response of polyblends for heating, the DSC measurements was also conducted. Appendix A represents the DSC thermograms of DGEBA/PEI and DGEBA/PEI/HRGO polyblends which confirms the HRGO accelerating the curing reaction process.

## 4. Conclusions

In summary, we developed a three-dimensional double-percolation network of a DGEBA/PEI/HRGO ternary system using a low content of 0.5 wt.% HRGO with the assistance of CRIPS. The results confirmed our prediction that a unique ternary nanocomposite can be fabricated by controlling the location of the conductive filler HRGO at the interphase. Furthermore, the electrical conductivity of polyblends increased by almost 16 orders of magnitude at a low content of 0.5 wt.% HRGO. The dynamic mechanical analyses demonstrated that the storage modulus was continuously enhanced by increasing PEI content and was enhanced by 15.4% with addition of 0.5 wt.% HRGO. The glass transition temperature (T_g_) also increased with addition of PEI. Therefore, the DGEBA/PEI/HRGO nanocomposites have significant potential for various engineering applications such as electronic packaging, EMI shielding, electrostatic discharge, etc.

## Figures and Tables

**Figure 1 nanomaterials-11-00228-f001:**
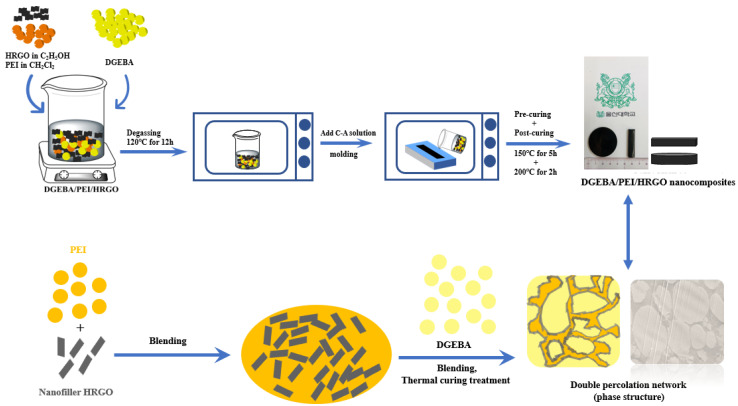
Schematic illustration of preparation of DGEBA/PEI/HRGO polyblend nanocomposites and the phase structure during the process.

**Figure 2 nanomaterials-11-00228-f002:**
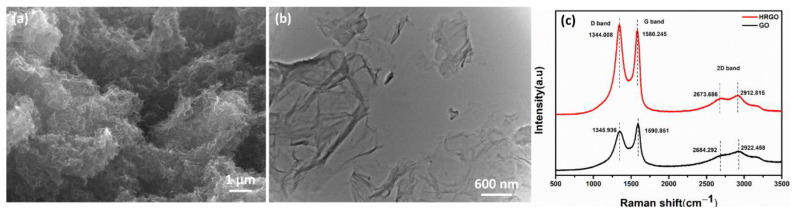
(**a**) SEM micrograph of HRGO flakes after drying, (**b**) TEM micrograph of HRGO and (**c**) Raman spectra of GO and HRGO flakes.

**Figure 3 nanomaterials-11-00228-f003:**
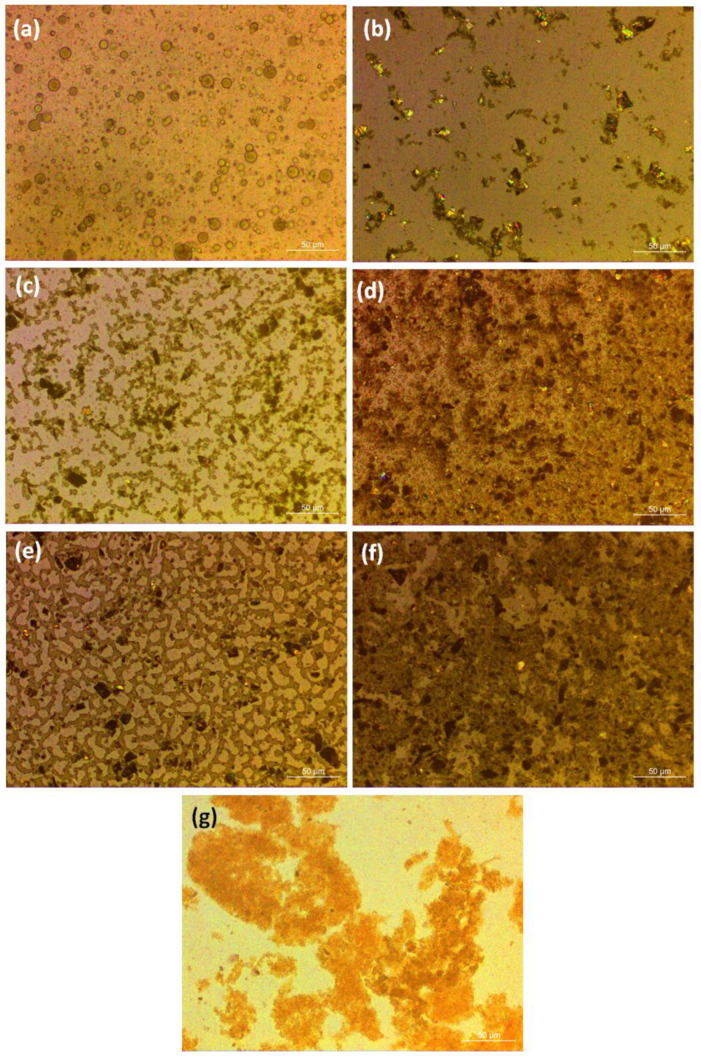
The final phase morphologies of DGEBA/PEI/HRGO nanocomposites are (**a**) DP5H, (**b**) DP10H, (**c**) DP15H, (**d**) DP20H, (**e**) DP25H, (**f**) DP30H, and (**g**) DP30, respectively.

**Figure 4 nanomaterials-11-00228-f004:**
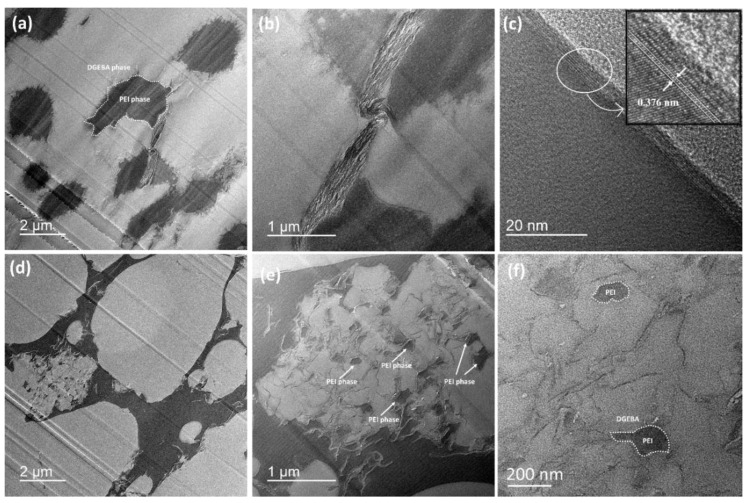
FETEM micrographs of (**a**–**c**) DP25H polyblend from lower to higher magnification with inset in (**c**) represents HRTEM micrograph of localized HRGO at the interphase of DGEBA and PEI, and (**d**–**f**) represents DP30H polyblend with phase inversion.

**Figure 5 nanomaterials-11-00228-f005:**
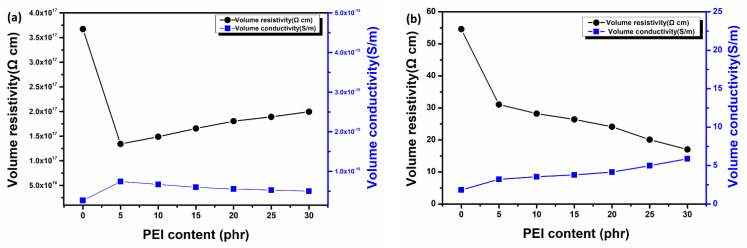
The electrical properties of samples for various PEI contents: (**a**) DGEBA/PEI, (**b**) DGEBA/PEI/0.5 wt.% HRGO.

**Figure 6 nanomaterials-11-00228-f006:**
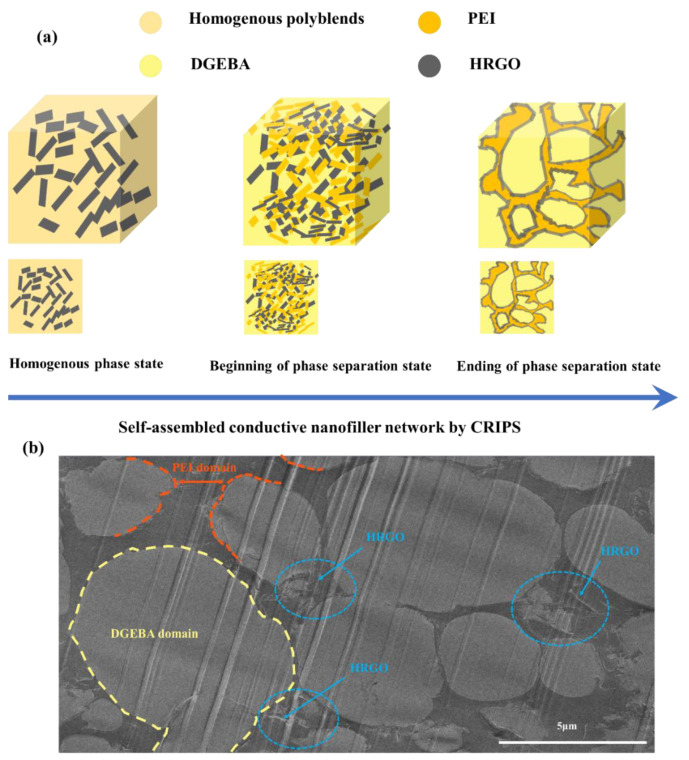
Schematic illustration: (**a**) structural evolution of double percolation conductive DGEBA/PEI/HRGO nanocomposites via CRIPS and (**b**) FETEM image of nanocomposite.

**Figure 7 nanomaterials-11-00228-f007:**
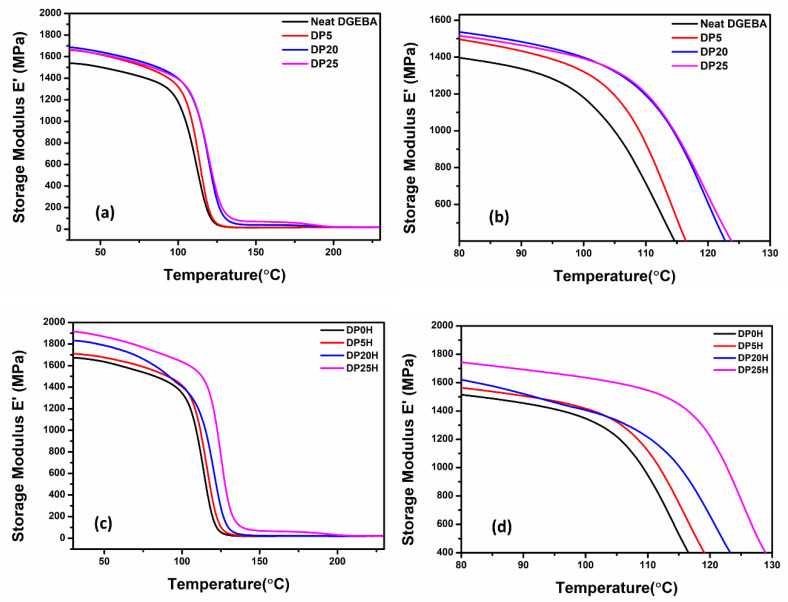
The storage modulus versus temperature of DGEBA/PEI polyblend (**a**) at 30–230 °C and (**b**) a magnified view of the beginning of the transition (80–130 °C). Storage modulus versus temperature curves for DGEBA/PEI/HRGO nanocomposites with 0.5 wt.% HRGO at (**c**) 30–230 °C and (**d**) 80–130 °C.

**Figure 8 nanomaterials-11-00228-f008:**
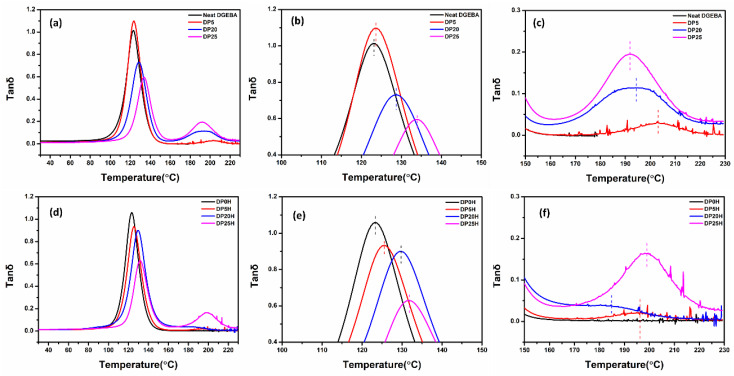
Tan δ versus temperature curve for DGEBA/PEI polyblend: (**a**) 30–230 °C, (**b**) 100–150 °C for T_g1_, (**c**) 150–230 °C for T_g2_, and tan δ versus temperature curve for DGEBA/PEI/HRGO nanocomposites: (**d**) 30–230 °C, (**e**) 100–150 °C for T_g1_, (**f**) 150–230 °C for T_g2_.

**Table 1 nanomaterials-11-00228-t001:** The volume resistivity and volume conductivity of DGEBA/PEI polyblend with and without HRGO.

Materials	Volume Resistivity (Ω cm)	Volume Conductivity (S/m)
Neat DGEBA	3.67 × 10^17^	2.67 × 10^−16^
DP0H	54.60	1.83
DP5H	31.06	3.22
DP10H	28.23	3.54
DP15H	26.41	3.78
DP20H	24.11	4.15
DP25H	20.09	4.98
DP30H	17.02	5.88

**Table 2 nanomaterials-11-00228-t002:** Comparison of volume resistivity and conductivity with those of previously published filler polyblend.

Samples	Polymer A	Polymer B	Filler Loading (wt. %)	Volume Resistivity (Ω cm)	Volume Conductivity (S/m)	[Ref]
EP/35PEI/GnPs	DGEBA	PEI	2	≈10^7^	≈10^−5^	[50]
EP/PES/MWCNT	DGEBA	PES	4.8	—	10^−1^	[36]
EP/PEI/CB	DGEBA	PEI	1	≈10^3^	—	[51]
DGEBA/PEI/MWCNTs	DGEBA	PEI	2	3.86 × 10^6^	—	[52]
DGEBA/PEI/AgNWs	DGEBA	PEI	3	9.6 × 10^5^ Ω (surface resistivity)	—	[53]
EP/AIN/MWCNTs	DGEBA	AIN	6	—	10^−10^	[54]
EP/PA/MWCNT	DGEBA	PA	1	—	10^−3^	[55]
DGEBA/PEI/HRGO	DGEBA	PEI	0.5	1.7 × 10^1^	5.88	This work

**Table 3 nanomaterials-11-00228-t003:** Surface tension components and parameters (in mJ/m^2^) and static contact angles for the test liquids DGEBA, PEI, and HEGO.

Liquid	γ_L_ (mJ/m^2^)	γ_L_^LW^(γ_L_^d^) (mJ/m^2^)	γ_L_^AB^(γ_L_^P^) (mJ/m^2^)	Contact Angle (θ)		
				DGEBA	PEI	HRGO
Deionized water (DI)	72.8	21.8	51	88.4	84.2	80.5
Glycerol (GL)	64.0	34.0	30	84.3	75.0	55.0
Formamide (FA)	58.0	39.0	0	54.1	58.8	27.1

**Table 4 nanomaterials-11-00228-t004:** The surface tension components and parameters (in mJ/m^2^) obtained from the contact angle data in Table 3 for the free surfaces of DGEBA, PEI, and HRGO.

Materials	γ (mJ/m^2^)	γ^d^ (mJ/m^2^)	γ^P^ (mJ/m^2^)
DGEBA	103.79	92.44	11.35
PEI	131.68	126.62	5.06
HRGO	93.99	93.67	0.32

HRGO: highly reduced graphene oxide; PEI: polyetherimide; γ = γ^d^ + γ^P^.

**Table 5 nanomaterials-11-00228-t005:** The wetting coefficient (ω_a_), interfacial tension (γ_pair_), and predicted localization of HRGO in DGEBA/PEI/HRGO polyblends.

Nanocomposites	Phase A	Phase B	Component Pair	γ_pair_ (mN/m)	ω_a_	Predicted Localization of HRGO
DGEBA/PEI/HRGO	DGEBA	PEI	DGEBA/PEI	7.74		
			DGEBA/HRGO	10.42	−1 < 0.171 < 1	Interphase
			PEI/HRGO	9.10		

**Table 6 nanomaterials-11-00228-t006:** The storage modulus, rubbery modulus, and T_g_ (from tan δ) of samples.

Materials	Storage Modulus E’ (MPa)	Rubbery Modulus (MPa)	Glass Transition Temperature (°C)	
			T_g1_	T_g2_
Neat DGEBA	1536	14.2	123.1	
DP5	1664	16.1	123.4	202.8
DP20	1687	38.8	128.8	194.1
DP25	1660	68.0	133.8	192.0
DP0H	1673	20.1	123.3	
DP5H	1709	22.4	125.6	196.5
DP20H	1831	23.5	129.7	184.7
DP25H	1915	64.8	131.6	198.2

Neat PEI’s T_g_: 217 °C.

## Data Availability

Data is contained within the article or Appendix A. Further, the published data can be reused by appropriate acknowledgement.

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
