# Peer review of "Construction and Mechanism Analysis of a Self-Assembled Conductive Network in DGEBA/PEI/HRGO Nanocomposites by Controlling Filler Selective Localization"

_nanomaterials, 2021, doi:10.3390/nano11010228_

Round 1

Reviewer 1 Report

Quite complex research, however, it requires very careful reading to follow all the processes and details. At first sight, in the version I downloaded from the journal, some errors in references are apparent ("Error! Reference source not found.") – lines 43, 47, 56, 77. Some references from the list at the end of the paper cannot be found in the text, which is a bit misleading. A number of formal inconsistencies in references in the text (missing spaces before the brackets, separate references [x], [z] vs. group [x] - [y]).

As I wrote before, the formulations of ideas are not always clear and require re-reading, e.g. line 72 (“was observed first presented by …”), line 280 “Therefore, Compared with Figure 4(a) and Figure 4(d) shows …”, line 296 “the results are opposite that shown…”, line 302 ”From the Figure 5(a) shows…”. Reintroducing acronyms several times is also distracting, the same as mistakes in grammar (third person singular, inconsistent use of tenses make the reader unsure what is a general statement and what is related to the present research). 

The paper would be shorter if the information was given only once, not repeated, e.g. Fig. 5 and Table 1, partly also Figure 8 and Table 5 gives the same data, half of the values in Table 2 are also given in the supplement, the same ideas are repeated in lines 315 and 394. Not very pleasant to read are also repeated words (e.g. in lines 310, 466 – 467).

I also noticed some mistakes in spelling (line 409 – verses, 431 – affect, inconsistent spelling: interface – interphase, in lines 144 vs. 161 – “rpm” vs “RPM”)

Beside these formal mistakes which distract the reader´s attention, there are also some more relevant unclear items, e.g.

  • in line 312 “After reaching 30 phr PEI” – however, the graph (Fig. 5) finishes at 30 phr, nothing about higher contents, so the behaviour above this concentration cannot be guessed.
  • In line 441 – is in Tg an increase by 10 °C really “drastic”?
  • Meaning of Figure 4 caption?

Probably the most substantial are the decrease in volume resistivity and the increase in volume conductivity by many orders, which are not appropriately commented in the text, only in Conclusion.

Please, pay attention to the readability of the paper.

Reviewer 2 Report

This manuscipt deserves publication in Nanomaterials journal having suitable topics. The paper is interesting and topical  developing  a three-dimensional double-percolation network of a ternary system DGEBA/PEI/HRGO  A value  of 0.5 wt.% HRGO represent its low content  and induces , the increase by almost 16 orders magnitude of the electrical conductivity of polyblends The results confirmed a prediction that a unique ternary nanocomposite can be fabricated by controlling the location of the conductive filler HRGO at the interphase. Before publication the paper needs moderate revision according to the following:

a) a better presentation of the paper novelty taking into account existing literature in the field

b) introducing DSC experiments not only for another  method for glass transition values, but as  a technique used to investigate the response of polymers to heating.

c)  values of table S3 need to be placed in the  paper and not  in supplementary materials

d)The conclusion "the DGEBA/PEI/HRGO nanocomposites  have significant potential for various engineering applications " is not clearly presented without some information  about applications

Reviewer 3 Report

The manuscript investigated the preparation and electrical properties of DGEBA/PEI/HRGO nanocomposites. Overall, the manuscript is well written and clear, and provides positive contributions to the field. This paper contains some new and useful information to be published in Nanomaterials. However before accepted for publication, the following improvement should be made.

  1. Although the language expression is clear, there still exist some minor grammatical, syntax or word usage errors in the manuscript. The English language should be carefully corrected.
  2. The result presentation and discussion could be more extensive, especially the electrical conductive mechanisms of the double-percolation DGEBA/PEI/HRGO nanocomposites. Before it can be published, some complementarities should be included.
  3. FTIR spectra of the HRGO should be provided.

Round 2

Reviewer 2 Report

 I do believe that the revised version is a better paper with significant improvement according to the comments and my recommendation for the new version is to be published in the present format in Nanomaterials journal

Reviewer 3 Report

The author has revised the manuscript based on the reviewer's comment. It might be accepted in the present form.